# UVB Irradiation-Induced Transcriptional Changes in Lignin- and Flavonoid Biosynthesis and Indole/Tryptophan-Auxin-Responsive Genes in Rice Seedlings

**DOI:** 10.3390/plants11121618

**Published:** 2022-06-20

**Authors:** Ga-Eun Kim, Me-Sun Kim, Jwakyung Sung

**Affiliations:** Department of Crop Science, Chungbuk National University, Cheongju 28644, Korea; lkn745@chungbuk.ac.kr (G.-E.K.); kimms0121@chungbuk.ac.kr (M.-S.K.)

**Keywords:** rice, UVB, lignin, flavonoid, indole/tryptophan, auxin

## Abstract

Global warming accelerates the destruction of the ozone layer, increasing the amount of UVB reaching the Earth’s surface, which in turn alters plant growth and development. The effects of UVB-induced alterations of plant secondary and cell wall metabolism were previously documented; however, there is little knowledge of its effects on rice seedlings during the developmental phase of leaves. In this study, we examined secondary metabolic responses to UVB stress using a transcriptomic approach, focusing on the biosynthetic pathways for lignin, flavonoid, and indole/tryptophan-auxin responses. As new leaves emerged, they were irradiated with UVB for 5 days (for 3 h/day^−1^). The genes encoding the enzymes related to lignin (4CL, CAD, and POD) and flavonoid biosynthesis (CHS, CHI, and FLS) were highly expressed on day 1 (younger leaves) and day 5 (older leaves) after UVB irradiation. The expression of the genes encoding the enzymes related to tryptophan biosynthesis (AS, PRT, PRAI, IGPS, and TS) increased on day 3 of UVB irradiation, and the level of tryptophan increased and showed the same temporal pattern of occurrence as the expression of the cognate gene. Interestingly, the genes encoding BBX4 and BBX11, negative regulators of UVB signaling, and SAUR27 and SAUR55, auxin response enzymes, were downregulated on day 3 of UVB irradiation. When these results are taken together, they suggest that secondary metabolic pathways in rice seedlings are influenced by the interaction between UVB irradiation and the leaf developmental stage. Thus, the strategies of protection against, adaptation to, and mitigation of UVB might be delicately regulated, and, in this context, our data provide valuable information to understand UVB-induced secondary metabolism in rice seedlings.

## 1. Introduction

The amount of ultraviolet B (UVB) radiation (280–320 nm) reaching the Earth’s surface has increased due to the depletion of the ozone layer, necessitating studies of the effect of UVB on crop plant growth and development [1]. Plants should absorb light energy beyond the capacity used in the light energy utilization process, i.e., photosynthesis, photorespiration, water cycle [2], and the xanthophyll cycle, to release heat [3]. These excessive energies generate reactive chemical species [2], causing oxidative damage, i.e., photodamage, negatively affecting growth [4,5,6].

Plants have developed different mechanisms to cope with excessive UVB (UVB stress). One strategy is to adjust their secondary metabolic pathways to promote the accumulation of protective metabolites as part of the defense response [7]. These metabolites include photosynthetic pigments such as chlorophyll and carotenoids, UVB-absorbing compounds such as flavonoids, phenolics, and alkaloids, and deposition of cuticular wax on the leaves [8,9,10]. A rapid conversion (activated) of UVR8, a photoreceptor that detects UVB, inhibits auxin biosynthesis which results in reduced plant growth [11,12,13].

Plant development age is an important determinant of stress susceptibility. Levels of antioxidants show a marked variation during the younger-to-older plant transition [14,15], and this transition is especially vulnerable to environmental stress [16]. Older leaves in monocotyledonous plants, including rice, physically enclose the younger leaves up until they grow to become the older ones [17]. This enclosure should protect younger leaves from more severe damage such as that induced by UVB. During the early vegetative stage, plants transfer sucrose from mature leaves to developing leaves via the phloem [18]. Taking these previous studies together, it is strongly required to understand how these key secondary metabolic processes are altered at the transcriptional level after supplemental UVB irradiation, especially in the leaf blades of rice seedlings undergoing developmental transition.

To validate our curiosity in relation to UVB-induced secondary metabolisms, we focused on the transcriptional changes in the secondary and cell wall metabolism-related genes and the indole/tryptophan-auxin response-associated genes, especially those that were upregulated as they may be part of the resistant and/or adaptive responses to UVB stress during the development of leaf blades. These data from RNA-Seq transcriptomic analysis were clearly confirmed by relative qRT-PCR and supported by assaying the relative levels of the cognate metabolites.

## 2. Results

### 2.1. RNA-Seq and De Novo Assembly of the O. sativa Transcriptome

The subset of genes activated and the levels of their cognate transcripts are partially dependent on rice response to the UVB dose used. In this experiment, the rice seedlings did not exhibit significant growth inhibition after UVB irradiation (Figure 1). The period of exposure led to dehydration by excessive photodamage, considering accompanying drought and heat stress [19].

Gene expression profiling by RNA-seq analysis was performed to better understand the potential metabolic changes based on the mRNA levels observed following UVB irradiation in rice seedlings. Plant samples were harvested at three different developmental points (days 1, 3, and 5 where the seedlings received cumulative doses of UVB irradiation at 3 h/day) since the emergence (1~2 cm-long) of new leaf blades. The expression level of genes was calculated by using the fragment per kilobase of exon model per million mapped reads (FPKM) value. DEGs analysis was performed using the leaf blades taken 1, 3, and 5 days after exposure to UVB (Figure 2).

The comparison of the genes differentially expressed between the control, i.e., with no exposure to UVB (-UVB), and the UVB-exposed samples (+UVB) showed a total of 3745 DEGs (2381 up- (63.6%) and 1364 downregulated (36.4%)) were identified to be changed by +UVB irradiation (Figure 3). The description and discussion of the results emphasize the upregulated genes.

### 2.2. Gene Ontology (GO) and KEGG Analysis

UVB irradiation has been shown to enhance secondary and/or cell wall organization metabolism [20]; therefore, Gene Ontology (GO) enrichment analyses were performed with 2381 of the upregulated DEGs by searching each UniProt_ID using the DAVID program. Plant response to UVB is also associated with the upregulation of plant secondary metabolic pathways such as the flavonoid and phenylpropanoid pathways and the accumulation of cell wall intermediates [20]. Thus, we pivoted our focus to explain the change in the secondary metabolic pathway that is activated by +UVB during the development of rice leaf blades. All DEGs can be divided by biological processes, and, of those, four subcategories were directly associated with secondary metabolic pathways: flavonoid biosynthetic process (32 genes), flavonoid glucuronidation (30 genes), plant-type cell wall organization (16 genes), and tryptophan biosynthetic process (six genes) (Appendix A). These 84 genes were assessed using Kyoto Encyclopedia of Genes and Genomes (KEGG) pathway analysis. As a result of the KEGG analysis, 11 genes encoding the enzymes directly involved in three metabolic pathways, lignin (three genes), flavonoid (three genes), and tryptophan biosynthesis (five genes), were finally selected and used to verify UVB-responsive metabolic changes using relative qRT-PCR.

### 2.3. Lignin and Flavonoid Biosynthesis Responses by RNA-Seq and qRT-PCR

The key functions of the selected genes encoding lignin (3)- and flavonoid (3)- biosynthetic enzymes were described (Appendix A). Lignin is derived from the phenylpropanoid pathway. It is one of the major compounds of the plant cell wall structure, which is important for cell wall fortification and is typically synthesized in older leaves rather than in younger ones [21]. Based on the RNA-seq results, six genes with a high Log_2_ FC (*p* < 0.05) were selected from the lignin and flavonoid biosynthesis pathways (Appendix A), and their expression was confirmed by qRT-PCR (Figure 4).

The transcriptional response of the selected genes of the lignin and flavonoid biosynthesis pathways after UVB irradiation and the levels of metabolites, i.e., of phenylalanine and tryptophan, determined by GC–TOFMS from our samples are depicted in Figure 4. The expression patterns of these genes from the younger-to-older transition of leaf blades during five consecutive days of UVB irradiation were observed by RNA-seq (Appendix A). The fold change (log_2_ scale) of each gene in +UVB/-UVB at day 1 was 3.9 (cinnamate-4-hydroxylase, 4CL), 2.7 (cinnamyl alcohol dehydrogenase, CAD), and 3.0 (peroxidase, POD) for lignin biosynthesis and 2.5 (chalcone synthase, CHS), 6.6 (chalcone isomerase, CHI), and 2.2 (flavonol synthase, FLS) for flavonoid metabolism (Appendix A). In contrast, the genes where there was no difference in expression due to UVB at days 3 and 5 were compared (Appendix A).

To validate the RNA-seq results, the relative expressions of the six genes were validated using qRT-PCR. The genes with a 0.02–6.6-fold higher expression under +UVB compared to −UVB were selected in the lignin and flavonoid metabolic pathways, and their expressions were compared at each of the three days (Figure 4). Except for cinnamate-4-hydroxylase (4CL) and flavonol synthase (FLS) at day 1, all the six genes were significantly higher expressed in response to UVB irradiation at days 1 and 5 (*p* < 0.05) (Figure 4). Additionally, UVB irradiation led to a significant increase in the relative levels of the cognate phenylalanine and tryptophan metabolites (Figure 4), which were 5.6- and 9.2-fold higher at day 3 and 6.6- and 8.1-fold higher at day 5, respectively.

### 2.4. Responses of Indole/Tryptophan Biosynthesis- and Auxin-Related Genes

The indole/tryptophan biosynthetic pathway is also known to be influenced by UVB irradiation [22], therefore, we compared the expression levels of the genes directly involved in indole/tryptophan biosynthesis by RNA-seq and qRT-PCR between −UVB and +UVB.

As mentioned earlier, the level of tryptophan, increasing 8.1~9.2-fold, showed large changes after the third day of UVB irradiation (Figure 5B). Based on the significant accumulation of tryptophan, we determined if there was a corresponding increase in the expression of some key genes encoding enzymes linked to the tryptophan biosynthetic pathway (Figure 5A,C). Five genes encoding the following enzymes were examined: anthranilate synthase (AS), anthranilate phosphoribosyl transferase (APRT), phosphoribosyl isomerase (PRAI), indole-3-glycerol phosphate synthase (IGPS), and tryptophan synthase (TS). To validate the results of the RNA-seq data (Appendix A), we also performed qRT-PCR of these genes, and similar trends were observed in the expression levels, although TS at day 3 did not differ (*p* > 0.05). Expression of all the five genes increased following increasing UVB irradiation, with the fold change indicated in parentheses—AS (3.4), APRT (3.0), PRAI (3.4), IGPS (3.3), and TS (1.4) at day 3, and AS (8.4), APRT (11.4), PRAI (3.9), IGPS (3.7), and TS (2.0) at day 5. Combining the RNA-seq and qRT-PCR results, the expression of the tryptophan biosynthesis-related genes was increased by UVB. 

Considering leaf curling by UVB irradiation (Figure 1) and the expression patterns of indole/tryptophan biosynthesis genes (Figure 5), one possibility was raised that auxin-related responses could be perturbed by UVB-triggered metabolic pathways. To understand this assumption, we assayed the expression patterns of the genes involved in auxin responses, i.e., *OsBBX4* and *OsBBX11*, a negative regulator of the UVR8–COP1 module, and *OsSAUR27* and *OsSAUR55*, an auxin-responsive gene. In RNA-seq, *OsBBX4, OsBBX11, OsSAUR27,* and *OsSAUR55* did not differ (Appendix A); however, the results of qRT-PCR tended to significantly decrease after treatment with UVB, except for *OsSAUR55* at day 3 and day 5 (Figure 6).

## 3. Discussion

The findings of this study and their implications are discussed in the broadest context possible, and the future research directions are also highlighted. UVB irradiation is an environmental factor that negatively affects many aspects of growth and development of economically important crops. Large numbers of studies have explored UVB-induced physiological [22,23,24,25] and morphological [26,27,28] perturbations. In this study, differences in the growth of rice seedlings were not observed between −UVB and +UVB. As described in the results, it was considered that the period of exposure and the intensity of radiation employed in this study was not sufficient to cause biomass reduction, although UVB irradiation tends to decrease rice seedling growth [25]. In conjunction with the global profiling of transcripts, proteins and metabolites have broadly been employed to advance our understanding of the complex metabolic networks that are activated in response to the stress derived from UVB irradiation in plants [22,29,30]. Despite abundant evidence that UVB induces secondary metabolism in plants, our understanding of the time-specific responses in developing rice leaves still remains limited. To address this question, we examined gene expression in rice leaf blades exposed to cumulative UVB irradiation (3 h/day^−1^ for 5 days) to uncover the associated transcriptional changes. UVB irradiation was initiated as new leaf blades emerged out of the sheath of the previous leaf (1~2 cm long) (Figure 1).

### 3.1. Lignin and Flavonoid Pathways against UVB Irradiation

Csepregi et al. [15] cautioned against pooling leaves of different development stages when trying to understand leaf UV responses. Our results revealed large differences in UVB exposure time between the lignin and flavonoid biosynthesis pathways. The level of phenylalanine, a precursor to lignin and flavonoid biosynthesis, was significantly increased starting day 3 after UVB irradiation compared to −UVB. The increase in aromatic amino acids, including phenylalanine, was categorized as “early responsive” metabolites, which were perturbed within hours of UVB exposure [22]. Kusano et al. [22] also demonstrated that carbon flux was dominant towards the shikimate pathway at the transcriptional level of the genes. Enhanced accumulation of secondary metabolites including lignin and flavonoids was proven as an early response against UVB stress [31,32,33,34,35] and was also observed in some plant species: rye [36], grapes [37], and cistus [38]. Our study carefully affords the probability that lignin and flavonoids feature both defensive and adaptive responses in an excessive UVB environment in terms of the increasing patterns in both younger (day 1) and older (day 5) stages and shows some accordance with previous observations.

CAD (cinnamyl alcohol dehydrogenase): up to 50% greater cell wall degradation in knockout plants compared to control [39,40,41,42]. Therefore, it can be predicted that plants with a high CAD gene expression have less cell wall degradation than control. In our experiment, the increased expression of CAD on day 5 is considered as a defensive mechanism to prevent UVB-induced cell wall degradation by adversely affecting epidermal cells as UVB was continuously exposed. Flavonoids are a strong UVB absorber, and anthocyanin is produced in leaf tissue to develop resistance to stress [43,44,45]. Anthocyanins are synthesized because they are more efficiently stored in the vacuoles due to their high solubility when irradiated with UVB [46,47]. In our experiment, the increased expression of CHS, CHI promoted the production of flavonones, precursors of flavonoids and anthocyanin. The difference between our experiment and those already reported in the literature is that FLS, which belongs to the flavonol synthesis pathway, was highly and predominantly expressed compared to the enzymes associated with the anthocyanin pathway. This means that flavonol, a strong absorbent, was selected rather than anthocyanin which is easy to store as a UVB absorber. It should be taken into account that, regardless of the UVB exposure time, we sampled growing leaves regardless of leaf maturity, whereas the UVB exposure experiments previously reported were from fully developed leaves. Compared to mature leaves, growing leaves are not sufficiently complete to produce energy, so it is predicted that they synthesized flavonols to immediately respond to accumulated UVB stress [48]. This showed the same trend as the previous experiment in *Sinapsis alba*, *Nasturtium officinal*, and *Spinacia oleracea* [49,50].

**Figure 6 plants-11-01618-f006:**
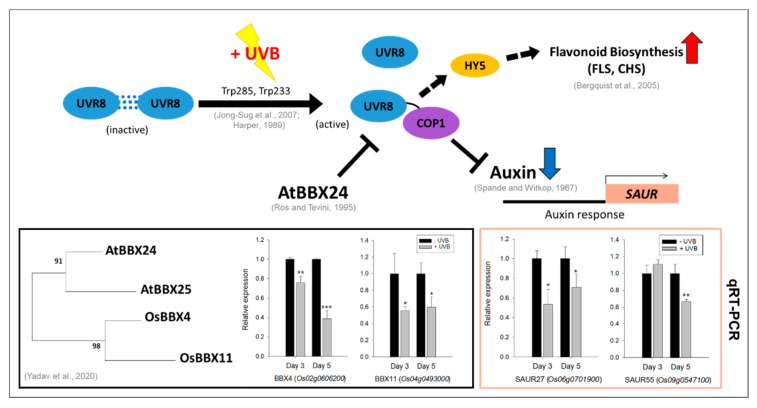
Expression of a negative regulator, *OsBBX4* and *OsBBX11*, and an auxin signaling-responsive gene, *OsSAUR27* and *OsSAUR55* (*n* = 6), and the proposed UVB-triggered metabolic processes (flavonoid and auxin responses) in developing rice leaf blades. Flavonoids and tryptophan were significantly enhanced by UVB irradiation. Four genes, *OsBBX4*, *OsBBX11*, *OsSAUR27,* and *OsSAUR55* were downregulated at days 3 and 5. Accordingly, it is assumed that auxin response is largely restricted by UVR8–COP1 and thus leads to the accumulation of tryptophan. Note: *, **, and *** mean *p* < 0.05, 0.01, and 0.001 (Student’s *t*-test) for qRT-PCR, respectively [47,48,50,51,52,53].

### 3.2. Responses of Indole/Tryptophan Biosynthesis and Auxin-Related Genes

To different degrees, all aromatic amino acids absorb ultraviolet light. Tryptophan is responsible for most of the absorbance of UV light (ca. 280 nm) [51]. It has been shown that UVB exposure interferes with indoleacetic acid (IAA) metabolism, which results in plant morphological changes due to hormonal imbalances [52], while, in contrast, UVB increases aromatic amino acids (AAAs) promoting the synthesis of compounds for protection [53]. This competitive diversion of flux from the IAA towards tryptophan biosynthesis promotes plant survival under UVB stress. Our previous study [25] showed that UVB irradiation resulted in the significant increase in tryptophan, a finding supported by other research groups [22,54,55,56]. In the current study, tryptophan increased dramatically after three consecutive days of UVB irradiation, and from this, a strong assumption can be made that the increase in tryptophan is greatly associated with the intensity of UVB irradiation and that the degree of development of rice leaves may also be important as a factor of UVB irradiation intensity in explaining increases in this AAA. The drastic increase in the expression of the genes encoding the major enzymes of the indole/tryptophan pathway at days 3 and 5, i.e., of AS (anthranilate synthase), APRT (anthranilate phosphoribosyl transferase), PRAI (phosphor ribosyl anthranilate isomerase), IGPS (indole-3-glycerol phosphate synthase), and TS (tryptophan synthase), supports this assertion. The qRT-PCR data (Figure 5) were also in accordance with the expression patterns of the genes detected in the RNA-seq data (Appendix A). The upregulation of these genes regulating indole/tryptophan metabolism in response to UVB irradiation was previously observed in the reports on *Arabidopsis* [57], *Catharanthus roseus* [58], and *Clematis terniflora* [59]. Indole and tryptophan are precursors for indole alkaloids and IAA. As described above, shikimate is an indispensable metabolite for secondary metabolic processes that enable plants to cope with adverse environmental stresses, including UVB.

On the basis of our current findings and available literature, UVB is likely to lead to a marked enhancement of flavonoids and tryptophan, while resulting in an obvious decrease in auxin signaling (Figure 6). UVR8 is activated by dissociation from dimers to monomers [60,61] and immediately dimerizes with COP1, triggering a UVB-specific response as a positive regulator for HY5 [62]. UVB regulates flavonoid metabolism through HY5 [63] and at the same time causes repression of auxin-responsive genes [64]. This, in turn, initiates changes in auxin transport associated with epithelial responses, causing leaf curling. Due to the operation of UVR8–COP1, the inhibition of auxin biosynthesis and signaling responses ultimately leads to accumulation of tryptophan, a precursor of auxin. The *AtBBX* family (*AtBBX20, AtBBX21,* and *AtBBX22*) plays an important role as a negative regulator of the UVR8–COP1 module, which is activated by UVB [65]. With the clear evidence from the previous studies described above, accumulation of tryptophan in our results is carefully considered by the negative feedback due to limited auxin signaling by the downregulated BBX gene family [66] and/or protein degradation in rice leaf blades [25]. Moreover, our data verified that UVB significantly decreased in the expression levels of *OsBBX4* and *OsBBX11*, and thus led to an inactivation and/or limited activation of UVR8–COP1 via negative regulation. Our results also report for the first time that downregulation of *OsSAUR27* and *OsSAUR55* was observed after UVB irradiation (days 3 and 5), which preferentially restricts auxin signaling responses, and are in line with those obtained by Vandenbussche et al. [67], who confirmed that auxin-responsive genes (*AtSAUR23* and *AtSAUR27*) were greatly downregulated by UVB irradiation in *Arabidopsis* plants.

## 4. Materials and Methods

### 4.1. Plant Material and UVB Irradiation

To investigate the effect of UVB on rice seedlings, rice (*Oryza sativa* L. “Saechucheng”) seeds were soaked by mixing 2.5 mL seed sterilizer (Kimaen, Farm Hannong) with 5 L of water and sterilized for 48 h at room temperature. After sterilization, the seeds were washed with water and kept at 25 °C for 7 days to hasten germination. After germination, the seeds were sown in plastic containers (54 cm × 28 cm × 5 cm) filled with pearlite (New Pearl Shine No. 2; GFC Co., Hongseong, Korea), and then half-strength Hoagland nutrient solution was supplied every 3 to 4 days. At the first-leaf stage, the rice seedlings were transferred to a growth chamber (VS-91G09M-2600, Vision Scientific, Korea) with a 14/10 h photoperiod, 60% (*w/v*) relative humidity (RH), and 24/20 °C day/night temperature. Before the stress treatment, light irradiation using LEDs (149 μmol m^−2^s^−1^) was performed. When the plants reached the third-leaf stage, they were exposed to supplemental UVB irradiation (8.6 kJ m^−2^/day^−1^) for 5 days (3 h/day^−1^, from 10:00 to 13:00) by installing a UVB lamp (UVB fluorescent tubes, 310 ± 10 nm) in the growth chamber. The UVB lamp was covered with 0.1 mm cellulose diacetate film (Cadillac Plastic Co., Baltimore, MD, USA) to prevent transmission at wavelengths below 280 nm. Growing leaf blades (fourth leaf) from approximately 30 rice seedlings (10 seedlings × three replicates) were harvested at 1 PM on days 1, 3, and 5 after UVB irradiation, frozen in liquid nitrogen, and stored at −80 °C. For the control sample, the same part (fourth leaf) of the non-UVB irradiated rice seedlings (10 seedlings × three replicates) was harvested at the same time (days 1, 3, and 5, at 1 PM).

### 4.2. RNA Extraction, cDNA Synthesis, and Quantitative Real-Time PCR (qRT-PCR)

Total RNA was extracted from the leaf blades of rice seedlings at 1, 3, and 5 days after −UVB and +UVB treatment, respectively, using the TRIzol reagent (Invitrogen, Carlsbad, CA, USA) according to the manufacturer’s instructions. The purity and concentration of the extracted RNA were estimated using NanoDrop (Thermo Fisher Scientific, Madison, WI, USA) and checked on a 1.2% agarose gel. First-strand synthesis performed using a Maxime RT PreMix Kit with oligo(dT) primers was used for cDNA synthesis from 1 µg of total RNA. Quantitative RT-PCR was performed to analyze the relative gene expression of RNA using EvaGreen Q Master (LaboPass, Seoul, Korea) with a CFX Connect Optics Real-Time System (Bio-Rad, Hercules, CA, USA) according to the manufacturer’s instructions. Quantification method 2^−∆∆Ct^ was used, and the variation in expression was estimated using three biological replicates. The rice actin gene was used as the internal control to normalize the data. The PCR conditions consisted of an initial denaturation step at 95 °C for 25 s, followed by 60 cycles of denaturation at 95 °C for 20 s, and annealing (Appendix A, 40 s) and extension (72 °C, 30 s) at a melting temperature (Tm, °C) designated by each of the primer sets.

### 4.3. RNA-Seq Library Construction, Sequencing, and DEGs Analysis

Sequencing was conducted on an Illumina HiSeq 2500 sequencing platform (Biomarker Technology Co.), provided by a commercial service provider (Thergen Bio, Seongnam, Korea) [68]. Raw sequencing reads in the FASTQ format were first filtered for quality using FastQC [69]. The high-quality paired-end reads were mapped to rice genome IRGSP-1.0 using Burrows–Wheeler Aligner (BWA) for estimating insert fragment sizes and standard deviations [70] and TopHat2 for aligning paired-end reads to the complete reference genome [71]. Normalization of the gene length was performed using Cufflinks [72]. Expression levels of each transcript were expressed as the kilobases per million fragments (FPKM) values to identify the differentially expressed genes (DEGs) [73]. All DEGs were determined using Cufflinks v2.0.1 (http://cufflink.cbcb.umd.edu accessed on 15 January 2021) [72] and categorized to the Gene Ontology (GO) framework using DAVID bioinformatics resources (ver. 6.8.) (https://david.ncifcrf.gov/tools.jsp accessed on 1 January 2021) [74]. To analyze the potential functions of rice proteins, genes were searched for in the Rice Annotation Project (RAP) databases, including the National Center for Biotechnology Information (NCBI), UniProt, and Kyoto Encyclopedia of Genes and Genomes (KEGG) pathway databases.

### 4.4. Metabolite Analysis 

The extraction of polar metabolites from powdered samples (100 mg) was carried out by adding 1 mL of 2.5:1:1 (*v*/*v*/*v*) methanol:water:chloroform [75]. Ribitol (60 µL, 0.2 mg mL^−1^) was used as the internal standard (IS). Briefly, the polar-phased extracts were subjected to methoxime (MO) derivatization and trimethylsilyl (TMS) etherification. The derivatized samples (1 µl) were quantified using an Agilent 7890A gas chromatograph (Agilent, Atlanta, GA, USA) coupled to a Pegasus high-throughput time-of-flight (HT TOF) mass spectrometer (LECO, St. Joseph, MI, USA). The quantitation of all the analytes was based on the peak area ratio of each analyte relative to the peak area of the IS. See the Appendix A for details.

## 5. Conclusions

The current study tried to advance our knowledge of the secondary metabolism as a plant defense system against cumulative UVB stress in rice seedlings using a transcriptomic approach. The expression patterns of the genes closely associated with lignin and flavonoid biosynthesis and indole/tryptophan-auxin response were validated with relative qRT-PCR at three timepoints (days 1, 3, and 5). The data from this study revealed that lignin and flavonoid biosynthesis and indole/tryptophan-auxin-responsive genes were largely dependent on UVB intensity- and/or leaf developmental stage. In terms of gene expression levels, lignin and flavonoids simultaneously played as plant defensive- (Day 1) and adaptative- (Day 5) roles, indole/tryptophan-auxin responses were induced after a certain intensity of UVB irradiation (Day 3~). Conclusively, this work provides the advanced knowledge that both secondary metabolisms in rice seedlings were closely dependent on the degree of an intensity of UVB irradiation with leaf developmental stages. From our results, further research raises to understand the variations in flavonoids and auxin responses by an activation/inactivation of UVB-triggered UVR8-COP1 module in different leaf development age.

## Figures and Tables

**Figure 1 plants-11-01618-f001:**
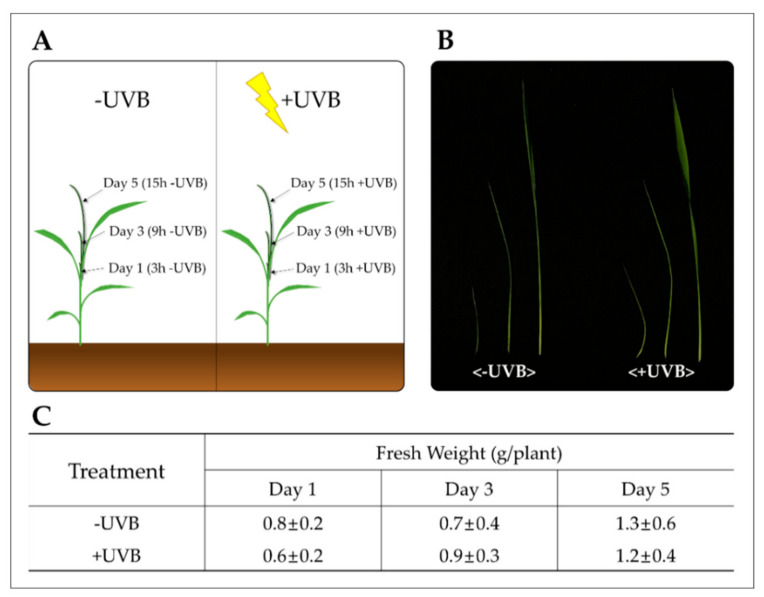
Development of leaf blades of rice seedlings: (**A**) growth of a leaf blade during the younger-to-older transition under −UVB (left) or +UVB (right); (**B**) fresh weight at 1, 3, and 5 days in both groups (*n* = 5). (**C**) UVB irradiation took place at 30~35 cm from the canopy for five consecutive days (3 h/day^−1^) as the new leaf blade (1~2 cm long) emerged from the previous leaf sheath.

**Figure 2 plants-11-01618-f002:**
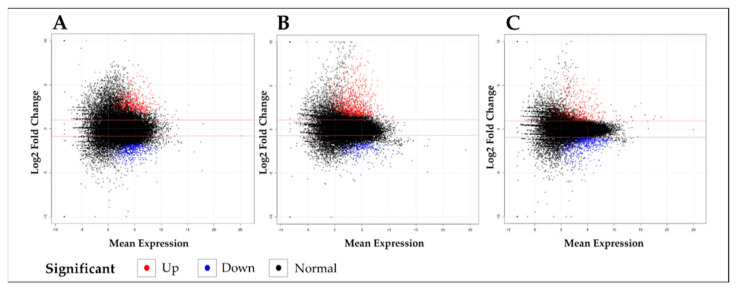
Scatterplots showing log_2_ fold change in gene expression in rice leaf blades after exposure to UVB compared to the control, i.e., day 0 untreated samples. The graphs show the tissues sampled at (**A**) day 1, (**B**) day 3, and (**C**) day 5, respectively. The red dots indicate the transcripts higher in the treated samples (log_2_ FC > 1; *p*-value < 0.05) and the blue dots indicate those that were lower (log_2_ FC < −1; *p*-value < 0.05) compared to day 0.

**Figure 3 plants-11-01618-f003:**
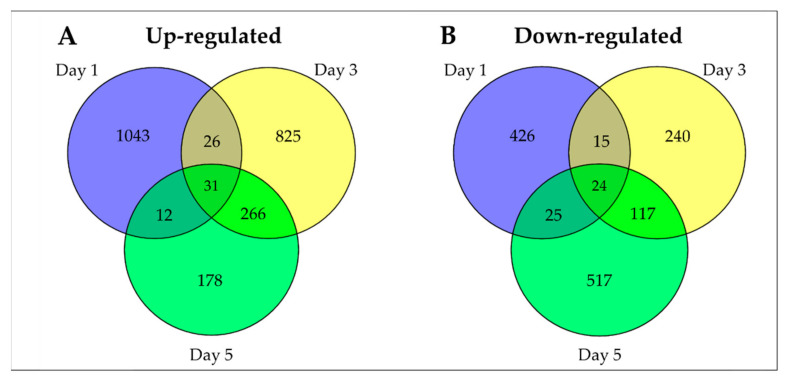
The Venn diagram showing the overlapping upregulated differentially expressed genes (DEGs) (**A**) and downregulated DEGs (**B**) in +UVB compared to −UVB during leaf development of rice seedlings.

**Figure 4 plants-11-01618-f004:**
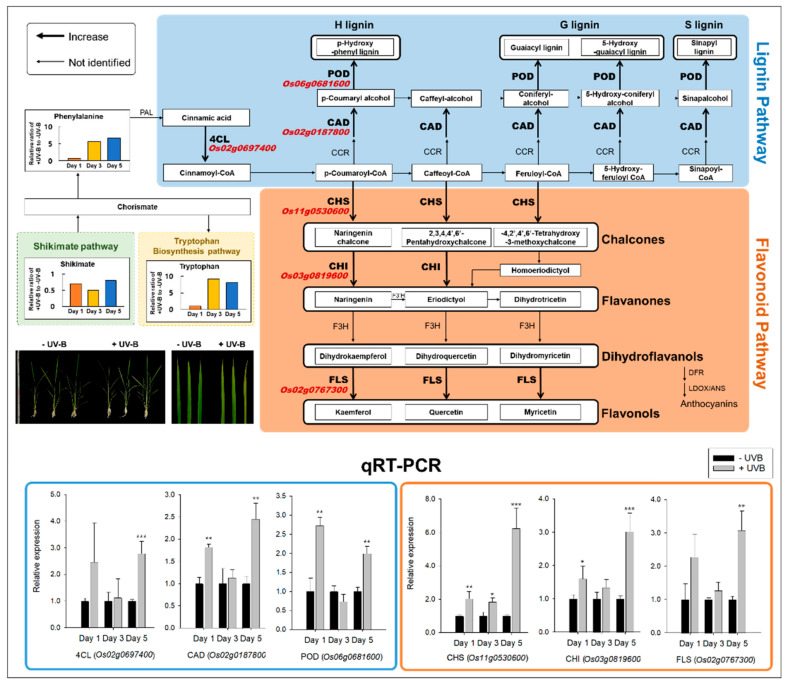
Variations of the key genes and metabolites employed in the metabolic process from phenylalanine to lignin and flavonoid biosynthesis. The bold black arrows indicate the genes that were upregulated in leaf blades under +UVB. Six genes were validated by qRT-PCR (mean ± SD, *n* = 6), and the bars in black and grey indicate −UVB and +UVB, respectively. Note: *, **, and *** mean *p* < 0.05, 0.01, and 0.001 (Student’s *t*-test) for qRT-PCR, respectively. Actin was used as the control. Acronyms: 4CL, cinnamate 4-hydroxylase; CAD, cinnamyl alcohol dehydrogenase; CCR, cinnamoyl-CoA reductase; CHI, chalcone isomerase; CHS, chalcone synthase; DFR, dihydroflavonol reductase; F3H, flavonone-3-hydroxylase; FLS, flavonol synthase; LDOX, leucoanthocyanidin dioxygenase; PAL, phenylalanine ammonia lyase; POD, peroxidase; qRT-PCR, relative quantitative real-time polymerase chain reaction.

**Figure 5 plants-11-01618-f005:**
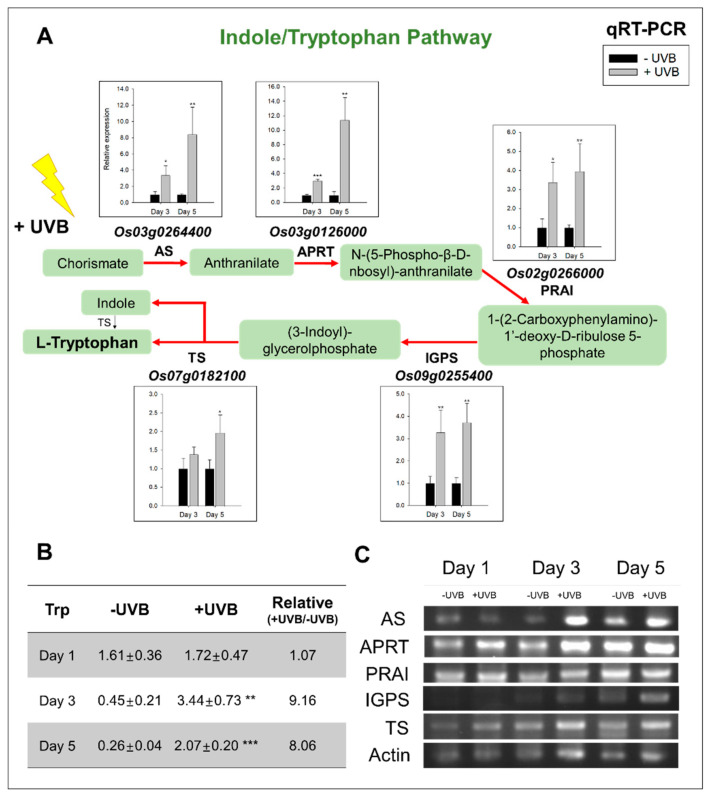
(**A**) Relative expression levels (+UVB/−UVB) of tryptophan biosynthesis (**A**,**C**)-involved genes by qRT-PCR (*n* = 6) and (**B**) the level of tryptophan (*n* = 3, peak area by GC–TOFMS) at three timepoints after UVB irradiation. The bars (**A**) in black and grey indicate −UVB and +UVB, respectively. Note: *, **, and *** mean *p* < 0.05, 0.01, and 0.001 (Student’s *t*-test) for qRT-PCR, respectively. Actin was used as the control. AS, anthranilate synthase; APRT, anthranilate phosphoribosyl transferase, PRAI, phosphor ribosyl anthranilate isomerase; IGPS, indole-3-glycerol phosphate synthase; TS, tryptophan synthase; qRT-PCR, relative quantitative real-time polymerase chain reaction.

## Data Availability

The datasets analyzed during the current study are available from the corresponding author upon reasonable request.

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
