# Peer review of "UVB Irradiation-Induced Transcriptional Changes in Lignin- and Flavonoid Biosynthesis and Indole/Tryptophan-Auxin-Responsive Genes in Rice Seedlings"

_plants, 2022, doi:10.3390/plants11121618_

Round 1
Reviewer 1 Report
The manuscript by Kim et al., expands our knowledge on the UV-B-induced changes in the gene expression of lignin, flavonoid biosynthetic and indole/tryptophan-auxin-responsive genes. The study is conducted in rice taking into consideration the three different duration of UV-B treatment, and thereby increasing dose of UV-B radiation and different developmental stage of the rice plant.
General comments:
From the methods, it seems the authors have compared 1d, 3d and 5d of UV-B treated plants with 0d of UV-B treated plants. Does this mean that the different timepoints samples were harvested when the plants were of different age? In this case it is difficult to say whether the differences in the response is due to the developmental stage of the plant or the cumulative UV-B dose.
This could have been avoided by a different experimental design where plants were harvested at the same time point, and UV-B treatment for 5 d could have been started earlier than 3 d, and 3 d earlier than 1 d. Alternatively, to have the non-UV treated plants also for 1d, 3d, and 5d as controls (at least for the qRT-PCR experiments).
Also, what was the white light irradiance that was used to grow the plants?
The overall discussion reads well, discussing broadly the results, but there is a bit of repetition from the results which I will highlight in the next section.
Specific comments:
Figure 1 legend: Panel B and C are interchanged.
Line 105 to 108: There were other GO terms which were also highly enriched in addition to those reported by the authors. Therefore it would be more correct to say that they focused on the relevant GO terms, instead of saying that they focused on the reported GO terms because those were the highest enriched.
Line 105 to 115: In addition, I was also a bit confused between the GO term and the KEGG pathway, as there was only GO:biological process figure in the supplemental. This part could be written more clearly.
Line 191 to 192: delete “except for the OsSAUR55 at day 3 and day 5”.
Line 211 to 215: reads more like results.
Author Response
"Please see the attachment."
We are sincerely grateful you for your time and consideration on our manuscript, “UVB Irradiation-Induced Transcriptional Changes in Lignin, Flavonoids Biosynthetic- and Indole/Tryptophan-Auxin Responsive-Genes in Rice Seedlings” to Plants, (Plant-1689596). Thus, it is with great pleasure that we resubmit our article for further consideration. We have incorporated changes that reflect that detailed suggestions you have graciously provided. We also hope that our edits and the responses we provide below satisfactorily address all the issues and concerns of editor and the reviewers have noted.
The original referee comments are provided in black color, whereas our answer are given in blue. The appropriate changes made in the revised manuscript are highlighted.
The main issue was whether or not it was necessary to consider the growth of the leaves to analysis the gene expression according to the UVB exposure time. To solve this problem, we specifically presented the experimental design and a clearer interpretation of the results.
We look forward to hearing from you in due time regarding our submission and to respond to any further questions and comments you may have.
Sincerely,
Ga-Eun Kim

Reviewer 2 Report
This paper describes the acquisition of a large amount of transcriptomic and metabolomic data. In papers like this, with large amount of data, it is important to clearly summarise the key findings arising from the data sets.
The authors need to clarify the influence that leaf development has on the expression of their target genes in order to claim the alteration of gene expression purely from increased UVB irradiation.
There are many reports on the regulation of the target pathways in many different plant species in response to UV irradiation, including the role transcription factors play in this process (both as promoters and repressors of gene expression). The authors need to highlight what findings they have made that are unique in this system compared to what is already in the literature. In addition, information on the expression of associated transcription factors is also necessary.
The authors state that they are only reporting on the up-regulation of gene expression. Down-regulated genes are also important in the the UV response. These need to be included in the analysis.
Author Response

(The authors gave the same response as above.)
